# Intracardiac Echocardiography Guidance Improves Procedural Outcomes in Patients Undergoing Cavotricuspidal Isthmus Ablation for Typical Atrial Flutter

**DOI:** 10.3390/jcm12196277

**Published:** 2023-09-29

**Authors:** Marton Turcsan, Kristof-Ferenc Janosi, Dorottya Debreceni, Daniel Toth, Botond Bocz, Tamas Simor, Peter Kupo

**Affiliations:** Heart Institute, Medical School, University of Pecs, Ifjusag utja 13, H-7624 Pecs, Hungary; marcittm@gmail.com (M.T.); janosikristof32@gmail.com (K.-F.J.); debreceni.d@gmail.com (D.D.); dani.toth0319@gmail.com (D.T.); boczbotond@gmail.com (B.B.); tsimor@hotmail.com (T.S.)

**Keywords:** intracardiac echocardiography, cavotricuspidal isthmus, CTI ablation, atrial flutter, ablation

## Abstract

Atrial flutter (AFL) represents a prevalent variant of supraventricular tachycardia, distinguished by a macro-reentrant pathway encompassing the cavotricuspid isthmus (CTI). Radiofrequency (RF) catheter ablation stands as the favored therapeutic modality for managing recurring CTI-dependent AFL. Intracardiac echocardiography (ICE) has been proposed as a method to reduce radiation exposure during CTI ablation. This study aims to comprehensively compare procedural parameters between ICE-guided CTI ablation and fluoroscopy-only procedures. A total of 370 consecutive patients were enrolled in our single-center retrospective study. In 151 patients, procedures were performed using fluoroscopy guidance only, while 219 patients underwent ICE-guided CTI ablation. ICE guidance significantly reduced fluoroscopy time (73 (36; 175) s vs. 900 (566; 1179) s; *p* < 0.001), fluoroscopy dose (2.45 (0.6; 5.1) mGy vs. 40.5 (25.7; 62.9) mGy; *p* < 0.001), and total procedure time (70 (52; 90) min vs. 87.5 (60; 102.5) min; *p* < 0.001). Total ablation time (657 (412; 981) s vs. 910 (616; 1367) s; *p* < 0.001) and the time from the first to last ablation (20 (11; 36) min vs. 40 (25; 55) min; *p* < 0.01) were also significantly shorter in the ICE-guided group. Acute success rate was 100% in both groups, and no major complications occurred in either group. ICE-guided CTI ablation in patients with AFL resulted in shorter procedure times, reduced fluoroscopy exposure, and decreased ablation times, compared to the standard fluoroscopy-only approach.

## 1. Introduction

Atrial flutter (AFL) is a common form of supraventricular tachycardia distinguished by the presence of a macro-reentrant circuit encompassing the cavotricuspidal isthmus (CTI), which denotes a narrow segment of tissue linking the tricuspid valve and the inferior vena cava [1].

For patients experiencing recurrent and symptomatic CTI-dependent AFL, the preferred initial therapeutic approach is radiofrequency (RF) catheter ablation. This procedure aims to establish a bidirectional conduction block across the CTI and exhibits a high degree of success in both short-term and long-term outcomes, with minimal occurrence of complications [2]. Despite the generally favorable efficacy of catheter ablation for CTI, it can be particularly challenging in some cases, frequently attributed to anatomical reasons [3].

Conventionally, electrophysiology procedures have been routinely conducted by employing fluoroscopy guidance, which exposes both patients and medical personnel to potentially hazardous doses of ionizing radiation [4]. In the context of catheter ablation procedures, intracardiac echocardiography (ICE) serves as a unique imaging technique that facilitates the immediate visualization of intracardiac structures in real-time (Figure 1).

Prior randomized clinical trials demonstrated that the utilization of ICE for CTI ablations significantly decreases radiation exposure in comparison with the fluoroscopy-only procedure. However, the data derived from the studies exhibited inconsistent findings concerning the effects of ICE-guided CTI ablation on procedural time and ablation time [5,6].

Therefore, the objective of our study was to comprehensively compare procedural parameters between the group undergoing CTI ablation guided by ICE and the group guided exclusively by fluoroscopy.

## 2. Materials and Methods

### 2.1. Study Population

In our retrospective, single-center study, we enrolled 370 patients who had undergone RF CTI ablation for either ongoing or documented typical AFL at our university hospital from January 2016 to January 2023. Patients included were divided into two groups based on whether ICE had been used during the procedure (ICE group and No ICE group), and the groups were compared on this basis.

We excluded patients referred for a second (redo) procedure, those who had previously undergone atrial fibrillation (AF) ablation or cardiac surgery, and individuals on whom procedures other than CTI ablation had been performed, due to different arrhythmias. We also excluded crossover cases in which ICE was not initially employed at the beginning of the procedure but was subsequently introduced following unsuccessful achievement of the procedural endpoint.

The study protocol adhered to the principles of the Declaration of Helsinki. Consent was obtained in accordance with the ethics research board of our institution. All patients underwent a baseline clinical assessment that encompassed their medical history, electrocardiography (ECG), routine blood tests, and echocardiogram.

### 2.2. CTI Ablation Procedure

The electrophysiological study and catheter ablation procedures were carried out under conscious sedation using midazolam and fentanyl, with patients in a fasting state and maintained on an uninterrupted anticoagulation regimen. The procedures were conducted by four skilled electrophysiologists who possessed a restricted background in ICE-guided CTI ablation techniques but held considerable expertise in performing fluoroscopy-guided CTI ablations. Subsequent to local anesthesia, following femoral venous access, a decapolar steerable catheter with an interelectrode spacing of 2-5-2 mm (Dynamic Deca, Bard Electrophysiology, Lowell, MA, USA) was positioned within the coronary sinus (CS). Additionally, a 7F irrigated 4 mm tip ablation catheter was inserted into the right atrium (Alcath Black Flux G, Biotronik, Berlin, Germany). The utilization of the 8F ICE catheter (AcuNaV™ 90 cm, Siemens Medical Solutions, Mountain View, CA, USA) was determined by the operators’ discretion. The echocardiographic transducer was situated within the inferior right atrium, specifically at the 6 o’clock orientation, with the possibility of lateral adjustment or orientation towards the septum as required. This positioning of the imaging plane facilitated the observation of anatomical reference points within the inferior right atrium, encompassing structures such as the CTI, the CS ostium, the tricuspid valve, the right ventricle, and the Eustachian valve (Appendix A). In instances where ICE was employed, an extra femoral vein puncture was carried out at the initiation of the procedure. No electroanatomical mapping system was used during the procedures.

In cases of ongoing arrhythmias, entrainment mapping was executed to confirm the cavotricuspidal dependence of the flutters. After discontinuation of the arrhythmia through RF ablation, the procedure was finalized under CS stimulation. For individuals presenting documented typical AFL yet manifesting a normal sinus rhythm during the procedures, ablation was carried out during continuous proximal CS pacing. Twelve-lead electrocardiogram recordings and intracardiac electrograms were acquired and archived utilizing a digital recording system (CardioLab, GE Healthcare, Chicago, IL, USA), incorporating a band-pass filter spanning the frequency range of 30 to 500 Hz.

RF ablation was performed to create a linear lesion along the CTI, employing a point-by-point approach during the ablation process. The ablation was performed in a temperature-controlled manner, maintaining a target temperature of 43 °C. Power delivery was confined to 45 W, while irrigation was maintained at a rate of 15 mL/min. The procedural endpoint included three main components: cessation of the arrhythmia, establishment of the bidirectional isthmus block, and the achievement of a comprehensive line of block. This line was characterized by distinct local double potentials placed at considerable intervals along it, indicating an isoelectric line between two sharp potentials as a result of the CTI ablation.

Major complications were defined as pericardial effusion/tamponade or vascular complications (e.g., major hematomas requiring intervention or prolonged hospitalization, atriovenous fistulas, and pseudoaneurysm).

Acute success was defined as the persistence of a bidirectional conduction block along the CTI following a 20 min observation period. The procedure duration, measured in minutes, spanned from the initiation of the first femoral puncture to the withdrawal of the last venous sheath. Therapy duration (in minutes) was measured from the first to the last RF application. Fluoroscopy time, also measured in minutes, and radiation dose were systematically documented by the fluoroscopy system. The duration of ablation (expressed in seconds) was calculated and stored using the EP recording system.

### 2.3. Statistical Analysis

The assessment of data distribution characteristics was conducted through the utilization of the Shapiro–Wilk tests. All statistical examinations were carried out with a two-tailed approach, adhering to a significance threshold of *p* < 0.05. Continuous data sets were portrayed as either the mean ± standard deviation or as the median accompanied by the interquartile range, contingent upon the appropriateness of the representation. Conversely, categorical variables were depicted in terms of absolute quantities and corresponding percentages. For the purposes of comparisons, the chi-square test, *t* test, and Mann–Whitney U test were employed as deemed suitable. The executions of the statistical analyses were facilitated using SPSS 28 software (SPSS, Inc., Chicago, IL, USA).

## 3. Results

A cumulative cohort of 370 patients were included in the study. Among these, 151 cases underwent exclusively fluoroscopy-guided procedures (designated as the “No ICE group”), while an ICE-guided approach was employed for the remaining 219 patients (referred to as the “ICE group”). The study population comprised 293 male individuals (constituting 79.2% of the total). There were no substantial disparities observed in terms of baseline characteristics between the two groups, with the exception of a lower prevalence of previously diagnosed AF in the No ICE group (68 out of 151 patients (45.6%)), compared to the ICE group (74 out of 219 patients (33.8%), *p* = 0.02). Detailed baseline characteristics of the study population are delineated in Table 1.

In each instance, a successful bidirectional isthmus block was achieved after a waiting period of 20 min, resulting in a 100% acute success rate. Notably, procedural time displayed a significant decrease within the ICE-guided cohort (No ICE group: 87.5 (60; 102.5) min vs. ICE group: 70 (52; 90) min, *p* < 0.001). The incorporation of ICE guidance led to a substantial reduction in treatment duration (defined as the time between the first and last ablation), evident through a comparison of 40 (25; 55) minutes to 20 (11; 36) minutes (*p* < 0.001). ICE guidance exhibited a substantial association with marked reductions in fluoroscopy time (900 (566; 1179) s vs. 73 (36; 175) s, *p* < 0.001) and decreased exposure to fluoroscopy (40.5 (25.7; 62.9) mGy vs. 2.45 (0.6; 5.1) mGy, *p* < 0.001). Additionally, the total ablation time experienced a decrease in the ICE group (910 (616; 1367) s vs. 657 (412; 981) s, *p* < 0.001). Importantly, the study population did not encounter any major complications. A summary of the results can be found in Table 2.

## 4. Discussion

In our single-center, retrospective study, we conducted a comparative analysis of procedural data between two groups: ICE-guided vs. fluoroscopy-guided CTI ablations. Our findings did not reveal any statistically significant differences between the groups in terms of major complications and acute success rate. However, the ICE-guided group exhibited several notable advantages over the fluoroscopy-only group, including shorter procedural time, fluoroscopy exposure, total ablation time, and therapy duration.

Catheter ablation plays a significant role in the long-term treatment of AFL by helping to maintain sinus rhythm. The European Society of Cardiology (ESC) guidelines for the management of patients with supraventricular tachycardia, published in 2019, provide recommendations regarding catheter ablation for AFL. In accordance with these guidelines, catheter ablation is recommended for individuals who undergo symptomatic and recurrent episodes of CTI-dependent flutter. Additionally, the ESC guidelines propose that the consideration of CTI ablation after the initial episode of symptomatic typical AFL should be considered [2].

During catheter ablation of the CTI, the primary objective is to create a continuous ablation line along the CTI, leading to the achievement of bidirectional block. Bidirectional block signifies the complete interruption of electrical conduction in both directions across the CTI, effectively eliminating the reentry circuit responsible for typical AFL. Achieving bidirectional block is considered a suitable endpoint for CTI ablation procedures. The existing body of scientific literature consistently demonstrates the high acute and long-term success rates associated with CTI ablation. Notably, multiple studies have substantiated that CTI ablation achieves a long-term success rate exceeding 90% [2,7,8,9].

The complex anatomical characteristics of the CTI constitute the primary factor contributing to challenges or limitations in achieving a complete and bidirectional conduction block within this region [3,5]. The intricate anatomical features of the CTI encompass several factors, including the elongated isthmus, prominent Eustachian ridges, and the presence of pouches [10,11]. Numerous publications have established a correlation between these anatomical complexities and various procedural factors. Specifically, an association has been identified between these anatomical variations and increased procedure duration, heightened radiation exposure, and an increased number of RF ablations required during CTI ablation for typical AFL [10,12,13,14].

The utilization of ICE is highly beneficial for the real-time visualization of cardiac anatomical structures during ablation procedures, particularly in patients with complex anatomies [15]. This imaging modality provides valuable assistance in guiding interventions and addressing the challenges posed by intricate anatomical variations. The application of ICE enables healthcare professionals to obtain enhanced visualization of internal cardiac structures, facilitating accurate catheter navigation and precise placement during ablation procedures. Real-time feedback from ICE aids in identifying critical anatomical landmarks and ensuring optimal catheter positioning, thereby improving the efficacy and safety of the intervention [15]. In addition to its guidance capabilities, the incorporation of ICE has the advantage of reducing the reliance on fluoroscopy, thereby mitigating the need for excessive fluoroscopy exposure during electrophysiological interventions [3].

Previous clinical trials have demonstrated that the incorporation of ICE during CTI ablations yields a substantial reduction in radiation exposure, compared to fluoroscopy-only procedures. In a prospective study involving 102 patients scheduled for CTI ablation conducted by Bencsik et al., the use of ICE was evaluated as a guiding tool during the procedure. The study aimed to assess the impact of ICE on success rates, procedure time, ablation time, radiation exposure, and complications. The results demonstrated that the ICE-guided group (*n* = 50) exhibited a significantly shorter procedure time, fluoroscopy time, and time spent on RF ablation, compared to the fluoroscopy-only group (*n* = 52). Additionally, the ICE-guided group experienced significantly lower radiation exposure and delivered RF energy, compared to the fluoroscopy-only group. Furthermore, seven patients (13%) in the fluoroscopy-only group crossed over to the ICE-guidance group due to prolonged unsuccessful RF ablation, and all of them were successfully treated. The incidence of vascular complications and recurrences were similar between the two groups [5].

Herman et al. conducted a comparative study involving 79 patients undergoing CTI ablation for typical AFL, comparing the use of the ICE-guided approach versus the fluoroscopy-guided approach [6]. Consistent with the findings of Bencsik et al., the authors reported a reduction in fluoroscopy time associated with the utilization of ICE. Interestingly, two patients in the fluoroscopy-only group required crossover to the ICE-guided approach to achieve a bidirectional conduction block. However, it should be noted that the use of ICE resulted in a longer total procedure time, compared to the fluoroscopy-only method. The observed increase in total procedure time in the ICE-guided group, as reported by the authors, can be attributed to the inclusion of an additional vein puncture, compared to the fluoroscopy-only group. This additional puncture contributed to the overall duration of the procedure.

Consistent with prior trials, our retrospective analysis demonstrated a significant reduction in fluoroscopy time associated with the utilization of ICE. Interestingly, contrary to the findings of the study conducted by Herman et al. and in alignment with the results of Bencsik’s study, we also observed a decrease in procedure time in the ICE-guided group. These differences in procedure time may be attributed to variations in the definitions of procedure time. In our study and Bencsik’s trial, procedure time was defined as the duration from the initiation of the femoral vein puncture to the withdrawal of the last venous sheath. However, in Herman’s study, the total procedure time encompassed the duration until successful hemostasis, incorporating the additional vein puncture and time required for achieving proper hemostasis. The use of ICE in CTI ablation procedures requires an additional venous puncture, potentially increasing the risk of vascular complications, particularly when utilizing larger 11F sheaths. However, it is noteworthy that both previous studies and our analysis consistently reported no significant increase in vascular complications within the ICE group. These findings suggest that the use of ICE, despite the supplementary venous puncture, can be safely performed without a notable elevation in adverse vascular events. Moreover, it is important to highlight that current evidence suggests the significant impact of incorporating vascular ultrasound guidance on reducing the incidence of these complications, even in patients receiving uninterrupted oral anticoagulation therapy [16,17,18]. Based on our exclusion criteria, we excluded patients who underwent crossover to ICE. Crossover to ICE is typically applied when standard, only-fluoroscopy-guided CTI ablation fails. Consequently, some challenging cases from the original fluoroscopy group were excluded. However, only three patients were excluded based on this criterion, so it did not significantly affect our results.

In recent years, the adoption of electroanatomical mapping systems (EAMS) as an alternative visualization modality in electrophysiology procedures has gained increasing popularity. This trend stems from the potential benefits associated with EAMSs, including the ability to decrease procedural time and minimize or completely eliminate radiation exposure. Multiple studies have provided compelling evidence supporting the feasibility and safety of employing a near-zero or zero-fluoroscopy approach utilizing EAMSs during CTI ablations, including their successful application as an extension to pulmonary vein isolation procedures within a single session [19,20]. Furthermore, the introduction of visualizable steerable sheaths has brought about notable progress in reducing radiation exposure during catheter ablation procedures [21].

Nevertheless, EAMSs have certain limitations in directly visualizing intracardiac structures, rendering them less advantageous for catheter ablation procedures in patients with atypical cardiac anatomy. In contrast, ICE emerges as a real-time imaging modality that overcomes this limitation by providing direct visualization of intracardiac structures, aiding in catheter positioning, stability assessment, and monitoring of lesion formation.

The utilization of ICE in cardiac arrhythmia ablation procedures has demonstrated significant benefits. Studies have consistently reported a substantial reduction in fluoroscopy time, fluoroscopy dose, and overall procedure duration when ICE is incorporated, compared to procedures performed without ICE guidance [22,23]. This highlights the added value of ICE in improving procedural efficiency, reducing radiation exposure, and optimizing outcomes in catheter ablation procedures. Moreover, a recent study has provided evidence demonstrating the feasibility of performing zero-fluoroscopy CTI ablation procedures using only ICE, without the need for EAMS [24]. Larger multicenter trials can evaluate the role of ICE in zero-fluoroscopy CTI ablations, with and without EAMS guidance.

The primary limitation of employing an ICE-guided approach for CTI ablation is the associated incremental cost. However, when compared to the additional expenses involved in utilizing EAMSs, the cost of ICE implementation is found to be comparable or potentially even lower [25], especially if the utilization of reprocessed ICE catheters is permitted [26].

## 5. Limitations

Some limitations need to be acknowledged. Firstly, the study design is retrospective, which means that the data are being collected and analyzed after the events have occurred. This introduces the potential for selection bias, incomplete data, and difficulties in controlling for confounding variables that might influence the outcomes. Secondly, the study was conducted at a single center, which might limit the generalizability of the findings. Furthermore, the operators’ varying experience in using ICE may have had an impact on the results. Patient populations and procedural practices can vary significantly between different medical centers, which could impact the external validity of the study’s results. Thirdly, although there were no differences in major complications between the groups, due to the lack of data, minor complications (e.g., hematomas not requiring any intervention) could not be compared. Finally, the study primarily focuses on procedural parameters and does not provide information about long-term clinical outcomes.

## 6. Conclusions

The utilization of ICE shortened procedure time, reduced fluoroscopy exposure, and decreased ablation time in patients who underwent CTI ablation for typical AFL, compared to the standard fluoroscopy-only approach.

## Figures and Tables

**Figure 1 jcm-12-06277-f001:**
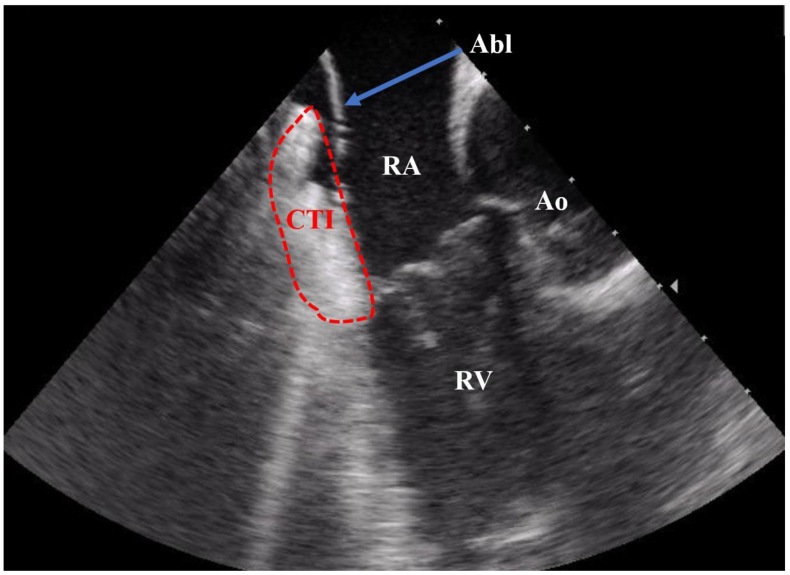
ICE recording during a CTI ablation procedure. Abbreviations: Abl = ablation catheter; CTI = cavotricuspid isthmus; RA = right atrium; RV = right ventricle; Ao = aortic root.

**Table 1 jcm-12-06277-t001:** Demographic and baseline characteristics of the study groups. Abbreviations: EF—ejection fraction; ICE—intracardiac echocardiography; LA—left atrium; TIA—transient ischemic attack.

	No ICE Group (*n* = 151)	ICE Group (*n* = 219)	*p* Value
**Age (years)**	65.4 ± 10.2	66.5 ± 10.1	0.31
**Male (%)**	115 (76.2)	178 (81.3)	0.23
**Hypertension (%)**	113 (74.8)	156 (71.2)	0.33
**Diabetes mellitus (%)**	48 (32.2)	61 (27.9)	0.37
**Reduced EF heart failure (%)**	36 (24.2)	52 (23.7)	0.93
**Coronary artery disease (%)**	49 (32.9)	61 (27.9)	0.30
**Chronic kidney disease (%)**	20 (13.6)	24 (11)	0.61
**Prior stroke/TIA (%)**	11 (7.4)	18 (8.3)	0.76
**Atrial fibrillation (%)**	68 (45.6)	74 (33.8)	0.02
**LA diameter (mm)**	56.3 ± 6	58 ± 7	0.16

**Table 2 jcm-12-06277-t002:** Procedural parameters in the study population. Abbreviation: ICE—intracardiac echocardiography.

	No ICE Group (*n* = 151)	ICE Group (*n* = 219)	*p* Value
**Total procedure time (min)**	87.5 (60; 102.5)	70 (52; 90)	<0.001
**Total ablation time (s)**	910 (616; 1367)	657 (412; 981)	<0.001
**Treatment duration (min)**	40 (25; 55)	20 (11; 36)	<0.001
**Total fluoroscopy time (s)**	900 (566; 1179)	73 (36; 175)	<0.001
**Total fluoroscopy dose (mGy)**	40.5 (25.7; 62.9)	2.45 (0.6; 5.1)	<0.001
**Acute success rate (%)**	100	100	1.0
**Major complication (%)**	0	0	NA

NA means not applicable.

## Data Availability

The data presented in this study are available upon request from the corresponding author. The data are not publicly available due to Hungarian legal regulations.

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
