# Peer review of "Intracardiac Echocardiography Guidance Improves Procedural Outcomes in Patients Undergoing Cavotricuspidal Isthmus Ablation for Typical Atrial Flutter"

_jcm, 2023, doi:10.3390/jcm12196277_

Round 1

Reviewer 1 Report

The paper is well written and understandable.

The reduction of procedural duration and fluoroscopy exposure is of interest even if electroanatomic mapping has a dominant role in this field.

In my opinion, there are two main issues:

1) exclusion criteria could have biased the study because the most difficult cases have been excluded. Could the authors comment this issue in the discussion?

2) the comparison with electroanatomic mapping is not a aim of this study; nevertheless, the authors acknowledged that this could be the ral gold standard. In my opinion, the discussion on this point is poor and reference 24 is wrong. 

Two minor issue:

In the discussion, authors underline the difference in procedural duration reported in other studies could be justified by the lone adjunctive puncture for ICE introduction.  Nevertheless, in this study is really lower. Could the authors comment this issue?

Furthermore, RF ablation could be performed witha 8 mm non irrigated ablator. Why have the authors chosen a different technique? Please, could the author comment this issue?

Author Response

The paper is well written and understandable.

The reduction of procedural duration and fluoroscopy exposure is of interest even if electroanatomic mapping has a dominant role in this field.

In my opinion, there are two main issues:

1) exclusion criteria could have biased the study because the most difficult cases have been excluded. Could the authors comment this issue in the discussion?

Thank You for your valuable comment. Regarding our exclusion criteria, we excluded procedures in which a crossover from fluoroscopy to ICE-guided was observed. We agree with the Reviewer that this might have excluded some of the most challenging CTI ablations in the original fluoroscopy group from our analysis. Upon reanalyzing our data, we found that crossover to ICE was used as an exclusion criterion in only three cases. As suggested, we have expanded the Discussion accordingly.

2) the comparison with electroanatomic mapping is not a aim of this study; nevertheless, the authors acknowledged that this could be the ral gold standard. In my opinion, the discussion on this point is poor and reference 24 is wrong. 

We also agree with the Reviewer that electroanatomical mapping system (EAMS)-guided CTI ablations are the standard worldwide to reduce or eliminate the need for fluoroscopy during procedures. In lines 249-258, we presented this information with relevant scientific literature. There is no scientific data regarding the role of ICE in cases of EAMS-guided CTI ablations. Luani et al., in a single-center study, showed that achieving zero fluoroscopy CTI ablations with only ICE guidance is feasible without the use of EAMS (Ref 24).

Two minor issue:

In the discussion, authors underline the difference in procedural duration reported in other studies could be justified by the lone adjunctive puncture for ICE introduction.  Nevertheless, in this study is really lower. Could the authors comment this issue?

Previously, two published trials aimed to evaluate the impact of ICE on procedural duration in patients with typical atrial flutter; however, the findings are inconsistent. In Bencsik's trial, the authors found significantly shorter procedural times in the ICE group, while in Herman's trial, significantly longer procedure durations were associated with the use of ICE. These authors explained the difference by an additional venous puncture, which required more time for hemostasis after sheath removal. As the Reviewer pointed out, we found shorter procedure times in the ICE group. Considering the different study designs, these findings cannot be directly compared. Our procedural time definition was the same as that applied in Bencsik's trial. The longer procedure time found in Herman's trial can theoretically be explained by either a different definition or a relatively long procedure time in the ICE-guided group, even though the procedure time was relatively short overall. Based on the above, the operator's experience with ICE can play a role in the effect of using ICE for CTI ablations.

We have added to the Limitations section that the experience of the operators in using ICE varied, which may have affected the results.

Furthermore, RF ablation could be performed witha 8 mm non irrigated ablator. Why have the authors chosen a different technique? Please, could the author comment this issue?

Throughout the entire study period, we did not have access to an 8 mm non-irrigated catheter at our center; therefore, all procedures for CTI ablation were performed using a 4 mm irrigated tip catheter, as mentioned in the Methods section.

 Once again, we would like to express our appreciation for your valuable comments.

Reviewer 2 Report

The authors wanted to compare procedural parameters between the group undergoing CTI ablation guided by ICE and the group guided exclusively by fluoroscopy.

In general the topic is interesting, the manuscript is well written. In my opinion routine usa of ICE in case of CTI ablation is too expensive, maybe as a bail-out in case of problems in fluoro-only strategy. Still it can be discussed in scientific context.

I have only few comments. First – some more precise information about major endpoint of the study should be provided – what was considered a CTI block? The term ‘placed at considerable intervals along the line’ is intuitively understandable for electrophysiologists, but not for all cardiologists (line 111).

Second - it would be very interesting if the authors had information about number of conduction recurrences during final 20 minutes of observation.

Third – one of the limitations is non-randomized design of the study.

The last technical issue – this was a retrospective study. Did all participants really sign an informed consent? Is this feasible?

Author Response

The authors wanted to compare procedural parameters between the group undergoing CTI ablation guided by ICE and the group guided exclusively by fluoroscopy.

In general the topic is interesting, the manuscript is well written. In my opinion routine usa of ICE in case of CTI ablation is too expensive, maybe as a bail-out in case of problems in fluoro-only strategy. Still it can be discussed in scientific context.

I have only few comments. First – some more precise information about major endpoint of the study should be provided – what was considered a CTI block? The term ‘placed at considerable intervals along the line’ is intuitively understandable for electrophysiologists, but not for all cardiologists (line 111).

We would like to express our gratitude to the Reviewer for dedicating their time to review our manuscript and for providing valuable comments. In response to the Reviewer's suggestion, we have rephrased the cited sentence to make it more understandable for general cardiologists.

Second - it would be very interesting if the authors had information about number of conduction recurrences during final 20 minutes of observation.

Unfortunately, we do not have access to this type of data.

Third – one of the limitations is non-randomized design of the study.

The last technical issue – this was a retrospective study. Did all participants really sign an informed consent? Is this feasible?

We also appreciate the Reviewer's question. Our institute is a university center, and under Hungarian legislation, when a patient provides consent for a proposed procedure by signing a consent form, they are also granting permission for their case to be used for scientific data processing, provided that appropriate ethical approvals are in place. Informed consent for invasive electrophysiology procedures was obtained from all the subjects involved in the study.

Reviewer 3 Report

The article is well written and is about a very interesting topic.

ICE is an evolving technique with promising applicabilities to the field of cardiac intervention, as the probes are improving in image resolution and 3D imaging.

Typical atrial flutter Cavo-tricuspid isthmus (CTI) ablation is already an overall very safe and successful procedure which makes it difficult to improve on.

The authors have shown that the use of ICE can shorten procedure, ablation and fluoroscopy time, although the latter probably not clinically relevant. 

As mentioned in the discussion, the use of ICE obviates the need for an additional femoral vein puncture, appropriate training and an ICE probe.

Congratulations for the authors for their work.

Would recommend expanding the discussion with an additional paragraph mentioning future directions in CTI ablation with ICE. Could this procedure ever be only echo-guided without the need for an fluoroscope?

Author Response

The article is well written and is about a very interesting topic.

ICE is an evolving technique with promising applicabilities to the field of cardiac intervention, as the probes are improving in image resolution and 3D imaging.

Typical atrial flutter Cavo-tricuspid isthmus (CTI) ablation is already an overall very safe and successful procedure which makes it difficult to improve on.

The authors have shown that the use of ICE can shorten procedure, ablation and fluoroscopy time, although the latter probably not clinically relevant. 

As mentioned in the discussion, the use of ICE obviates the need for an additional femoral vein puncture, appropriate training and an ICE probe.

Congratulations for the authors for their work.

Would recommend expanding the discussion with an additional paragraph mentioning future directions in CTI ablation with ICE. Could this procedure ever be only echo-guided without the need for an fluoroscope?

We would like to thank the Reviewer for dedicating time to review our manuscript and for providing thoughtful feedback. We appreciate the points raised regarding the safety and success of CTI ablation, and we are pleased that our study has shown potential benefits in terms of procedure, ablation, and fluoroscopy time with ICE guidance.

It's worth noting that Luani et al. conducted a study reporting that only ICE-guided CTI ablations are feasible and safe, even without the use of an electroanatomical mapping system. We have mentioned this in lines 270-273 (Ref 24).

As per your recommendation, we have expanded the Discussion section to include potential future trials that could evaluate the role of ICE in zero-fluoroscopy CTI ablations, both with and without electroanatomical mapping-system guidance.

Once again, we express our sincere appreciation for the valuable comments provided by the Reviewer.